# Multi-Resolution Analysis with Visualization to Determine Network Attack Patterns

Dong Hyun Jeong [1],*[ID], Bong-Keun Jeong [2][ID] and Soo-Yeon Ji [3],*[ID]

1   Department of Computer Science and Information Technology, University of the District of Columbia, Washington, DC 20759, USA
2   Department of Management and Decision Sciences, Coastal Carolina University, Conway, SC 29528, USA
3   Department of Computer Science, Bowie State University, Bowie, MD 20715, USA
*   Correspondence: djeong@udc.edu (D.H.J.); sji@bowiestate.edu (S.-Y.J.);
    Tel.: +1-202-274-6292 (D.H.J.); +1-301-860-4458 (S.-Y.J.)

**Abstract:** Analyzing network traffic activities is imperative in network security to detect attack patterns. Due to the complex nature of network traffic event activities caused by continuously changing computing environments and software applications, identifying the patterns is one of the challenging research topics. This study focuses on analyzing the effectiveness of integrating Multi-Resolution Analysis (MRA) and visualization in identifying the attack patterns of network traffic activities. In detail, a Discrete Wavelet Transform (DWT) is utilized to extract features from network traffic data and investigate their capability of identifying attacks. For extracting features, various sliding windows and step sizes are tested. Then, visualizations are generated to help users conduct interactive visual analyses to identify abnormal network traffic events. To determine optimal solutions for generating visualizations, an extensive evaluation with multiple intrusion detection datasets has been performed. In addition, classification analysis with three different classification algorithms is managed to understand the effectiveness of using the MRA with visualization. From the study, we generated multiple visualizations associated with various window and step sizes to emphasize the effectiveness of the proposed approach in differentiating normal and attack events by forming distinctive clusters. We also found that utilizing MRA with visualization advances network intrusion detection by generating clearly separated visual clusters.

**Keywords:** visualization; intrusion detection analysis; discrete wavelet transformation

## 1. Introduction

Protecting computing infrastructures from cyber threats is one of the priority research topics in network security. The complex nature of traffic data generated by recent technologies and software applications and constant changes in network traffic patterns make it difficult to detect threats and secure the network from unauthorized access and harm. Intrusion detection systems monitor incoming and outgoing packets in a network and detect any abnormal behaviors or violations of security policies. With the increasing amount of computer malware and malicious attacks becoming more sophisticated, the need for designing effective intrusion detection systems has increased significantly. Numerous intrusion detection techniques have been proposed, but improving the accuracy and efficiency of systems remains a research challenge. Traditional intrusion detection techniques discover intrusive activities or network attacks by analyzing packets at a network layer and comparing them to known attack signatures. However, this approach is not suitable for identifying unknown (or new) threats. Thus, an alternative approach has been proposed to analyze network traffic data by referencing normal network traffic patterns. In detail, rather than searching for known attacks, it utilizes a mechanism to train the system to learn normal network behavior (as a possible baseline). Any deviation from the baseline will trigger alerts. One major limitation of this approach is that it suffers from high false rates.

In this paper, we propose an approach to analyzing network traffic data by integrating multi-resolution analysis (MRA) and visualization. MRA [1] examines data to represent different scales or frequency components based on signal processing techniques. For MRA, wavelet transform is considered because it decomposes data into multiple lower-resolution levels and determines dominant modes of variability [2]. It has been widely used in analyzing non-stationary data to determine anomalies, including false network events or network intrusions [3–5]. Among various wavelet methods, such as Continuous Wavelet Transform (CWT), Fast Wavelet Transform (FWT), Discrete Wavelet Transform (DWT), and Stationary Wavelet Transform (SWT), DWT is utilized in this study to extract features from network traffic data. After determining the number of features based on input parameters, statistical validation is performed to evaluate the selected features and remove non-statistically significant elements. Principal Component Analysis (PCA) is then applied to determine the new vital variables from the selected features to support the analysis of network traffic activities in visualization. To highlight the effectiveness of our proposed method, we performed an in-depth analysis with publicly available intrusion detection datasets. The main contributions of our work are:

- We performed an in-depth analysis of MRA with multiple intrusion detection datasets.
- We evaluated various wavelet functions to find the optimal wavelet for analyzing network traffic activities utilizing visualization techniques.
- We also conducted classification and clustering analysis to detect network attack patterns.
- To the best of our knowledge, our work is the first visual analysis to understand the effectiveness of MRA with visualization in analyzing network traffic data.

The rest of the paper is organized as follows. First, we discuss prior research in network intrusion detection and the benefits of utilizing visualization techniques. Then, the MRA method is explained in greater detail with an emphasis on the importance of a visualization tool. In Section 6, we present the performance evaluation results and conclude with a discussion, conclusion, and future work.

## 2. Previous Work

The massive volume of network traffic data generated by Internet activities, as well as the evolution of malware, poses a significant challenge to network security. For this reason, designing an innovative intrusion detection system has gained significant attention and is considered one of the major research areas in cybersecurity. There are two primary approaches to intrusion detection system implementation: signature-based detection techniques and anomaly-based detection techniques [6]. Signature-based techniques (also known as knowledge-based or misused-based techniques) apply pattern-matching processes to detect known attacks or system vulnerabilities. The signature-based techniques compare current network traffic patterns with the previous intrusion signatures in a database. Therefore, they are effective in detecting known attacks with minimum false alarms [7]. Anomaly-based or behavior-based techniques monitor network traffic and compare it with normal or expected traffic profiles, including bandwidth usage, common protocols, combinations of port numbers, and system information. Any significant deviation from the baseline traffic pattern is considered an anomaly or intrusion [8]. Although the two types (anomaly vs. behavior) share considerable overlap and are often regarded as identical in the literature, there is a slight difference between the two. Anomaly-based techniques create a normal profile by training current network traffic and then using the profile to detect deviations. On the other hand, behavior-based techniques do not necessarily compare against the baseline profile.

Signature-based and anomaly-based techniques are effective in detecting network intrusions. However, there are several limitations and disadvantages as well. The signature-based techniques are generally less effective against unknown attacks or new deviations of similar attacks. Furthermore, the system needs a constant update with new attack signatures, which may require considerable resources and overhead. The anomaly-based

techniques show a significant performance issue (high false positive alarm) under heavy or sudden traffic bursts [9]. Additionally, building a normal network traffic profile is not easy when network systems and computing environments are complex and diverse. To address the limitations, researchers have proposed techniques that integrate statistics-based, knowledge-based, and machine learning-based techniques [10,11]. Statistic-based anomaly detection techniques aim to build a distribution model (univariate, multivariate, and time series) for a baseline network profile. Using statistical functions (e.g., median and standard deviation), low-probability events are identified and classified as possible anomalies. Knowledge-based anomaly detection techniques create a knowledge database based on normal network traffic patterns. Unlike statistic-based techniques, a set of rules to define baseline profiles is developed based on human or expert knowledge. Studies showed that knowledge-based techniques are particularly effective in reducing false-positive alarms [8]. Machine learning-based techniques identify and classify security threats using some mechanism. The main difference is that the systems learn and improve their ability to detect anomalies based on their experiences without being actually programmed [12]. Because of this advantage, machine learning-based techniques have been applied extensively to design intrusion detection systems. Machine learning based-techniques encompass data mining techniques such as fuzzy logic, bayesian networks, genetic algorithms, clustering, decision trees, neural network, and support vector machines.

Cannady [13] emphasized the usefulness of Artificial Neural Networks (ANNs) for intrusion detection. However, the author stated that the accuracy of detecting network intrusions is closely dependent on datasets and methods used in training. Furthermore, ANNs do not necessarily provide a detailed explanation/reason regarding detected intrusions. Amor et al. [14] utilized Naïve Bayes (NB) in intrusion detection. As a simplified Bayesian probability model, the NB classifier operates based on the likelihood that one attribute does not affect others. Their experimental study result indicated that NB was faster than Decision Tree (DT) in terms of learning and classifying, but no significant performance difference was found between the two techniques. As the number of studies on designing new machine learning-based intrusion detection systems increased in early 2000, Nguyen and Armitage [15] surveyed various network traffic classification algorithms. They found that different ML algorithms demonstrated high accuracies, such as AutoClass, Expectation Maximisation (EM), DT, and NB. However, most approaches are uniquely designed to define their classification models by evaluating different test datasets. As a result, the models tend to be less effective in analyzing different datasets and network circumstances.

Researchers have continuously sought and adopted various ML algorithms to improve intrusion detection. Wang [16] showed the effectiveness of logistic regression (LR) modeling to detect multi-attack types. Albayati and Issac [17] compared the performance among NB, Random Tree Classifier (rTree), and Random Forest Classifier (rForest), and found that rForest was superior to other methods while maintaining a low false alarm rate. Support Vector Machine (SVM) has been used in intrusion detection analysis by numerous researchers. SVM classifies the input data using a set of support vectors that represent data patterns. It is well-suited for data classification by finding the hyperplane that maximizes the margin among all intrusion classes [4,18–21]. However, SVM classification depends mainly on the applied kernel types and parameter settings [21]. It also requires longer training time compared to other classification algorithms. A study by Khan et al. [18] proposed an approach to integrate hierarchical clustering analysis to address the limitation of SVM. Genetic Algorithm (GA) has been applied in intrusion detection for optimization, automatic model generation, and classification [22,23]. GA is a search algorithm that utilizes the mechanics of natural selection and genetics. It is often used to generate detection rules or select appropriate features from input data. However, the classification accuracy using GA tends to be slightly lower than tree algorithms such as J4.8 and Classification and Regression Trees (CART) [24]. CART is an algorithm that generates a set of rules by splitting data into each child node. It predicts continuous dependent variables and categorical independent variables by building a tree [25,26]. Unlike many statistic-based techniques, CART does

not require any distribution assumptions. It also supports datasets with multiple data types and missing values. Despite the fact that classical PCA has a high sensitivity to outliers [27], PCA is often used to extract significant features from network traffic data. For instance, Xu and Wang proposed (2005) a hybrid intrusion detection model based on PCA and SVMs. They found that the proposed method presented superior classification performance and improved accuracy [28].

Although machine learning based-techniques improve the accuracy and performance in detecting network intrusions, there are several limitations as well. They include high complexity and computational time, lack of ability to support real-time detection, high false positive rate, slow detection rate when applied to a large amount of data, and performing efficiently only with a single dataset [29]. We propose an integration of multi-resolution analysis (MRA) and visualization to address some of the issues. MRA utilizes signal processing techniques (i.e., wavelet transform) to discover any pattern changes. One of the main advantages of MRA is that it offers simultaneous localization in time and frequency, which can be used to detect sudden bursts from the data [4]. By decomposing input data until a pre-determined level, it can separate different levels of information (fine details vs. high level). Furthermore, prior literature indicates that MRA is computationally fast and appropriate for noise filtering and data reduction [30,31]. In summary, numerous studies have been conducted to design effective network intrusion detection systems. However, it is important to note that one algorithm or approach cannot detect all existing or unknown attacks precisely due to the existence of anonymity in network traffic patterns.

In the visualization community, researchers started to utilize various visualization techniques to address the limitations of traditional network intrusion detection analysis. Due to the importance of analyzing complex network traffic data, the community provides network traffic data as a part of the visualization challenge and motivates researchers to get involved in the visual analysis of real-world network traffic data. For example, the VAST 2012 challenge [32] provided the data from a financial institution to identify anomalies or problems visually and understand the health of a global corporate network. Shiravi et al. [33] conducted a comprehensive review of existing network security visualization systems and classified them into five different use-case classes: host/server monitoring, internal/external monitoring, port activity, attack patterns, and routing behavior. While numerous network security visualization systems have been proposed, most systems focus only on how to represent collected log data or network events. To better understand network traffic patterns and detect network intrusions more accurately, visualization techniques should be integrated with computational and machine learning approaches. In the following sections, a detailed explanation regarding how we combine both computational approaches and visualization techniques is presented.

### 3. Datasets

Three publicly available intrusion detection datasets were used in this paper: NSL-KDD [34], Kyoto 2006+ dataset [35], and CIC-IDS2017 [36]. NSL-KDD is a cleaned dataset that addresses the duplicated entry problems in the KDD'99 dataset [37]. The dataset includes approximately 150,000 records with 41 attributes. Twenty-four attack scenarios are grouped into four attack categories: DoS, R2L, U2R, and Probe. DoS denotes a Denial-of-Service attack that represents the attempts to make computing or network resources inaccessible. R2L (Remote to User) sends packets to another computing machine over the network to gain access to local user accounts. U2R (User to Root) gains a normal user account and exploits some vulnerabilities to gain root access to computing systems. Probe scans a network of computers to gather information and find vulnerabilities in computing machines. In this study, U2R was not considered due to the limited sample size (only 119 instances).

The Kyoto 2006+ dataset was developed to respond to the need for generating new network intrusion datasets. Although the KDD'99 and NSL-KDD datasets have been popular in designing machine-learning-based intrusion detection systems, it does not reflect recent

network traffic trends. The Kyoto 2006+ dataset includes millions of real traffic data records with 24 attributes [35]. The 14 commonly used features in intrusion detection studies were included. They are duration (length of connection), service type, source bytes, destination bytes, count, same_ser_rate, serror_rate, srv_serror_rate, dst_host_count, dst_host_srv_count, dst_host_same_src_port_rate, dst_host_serror_rate, dst_host_srv_serror_rate, and flag. Additional 10 features were added to understand traffic patterns in the networks. The initial version of the dataset was collected for three years (2006–2009), and the data collection continued until 2015. Since the size of the Kyoto 2006+ dataset is extremely large, we used a portion of the recent dataset (January 2015). The CIC-IDS2017 dataset [36] was created at the Canadian Institute for Cybersecurity (CIC). The dataset captures current network patterns to address the limitations of existing intrusion detection datasets. The CIC-IDS2017 dataset was captured during the period of 3 to 7 July 2017. It contains about 2.8 million network events with about 0.5 million attack events. The dataset consists of eighty-five variables, including the original full packet payload dataset, the processed dataset with a network traffic flow analysis tool (called CICFlowMeter), and labeled network flows with time stamps, IP addresses, network ports, protocols, and attack information. We used the dataset generated on Tuesday, which includes 431,873 normal and 13,835 attack events. The attack events were generated from Brute Force attack scenarios. In the rest of this paper, we use the terms NSLKDD, Kyoto2006, and CICIDS2017 to indicate each dataset utilized in our study, respectively.

## 4. Multi Resolution Analysis (MRA)

MRA generates multiple levels of representation to understand network traffic data by utilizing signal processing techniques. Prior to analyzing the datasets, data pre-processing was applied to exclude categorical attributes and null values. Then, a Discrete Wavelet Transform (DWT) was used to discover underlying patterns in different resolution levels ($\gamma$). The DWT decomposes data into a set of mutually orthogonal wavelet basis functions with a function, $\psi$, known as "mother wavelet". The wavelet basis functions indicate dilated, translated, and scaled versions of the mother wavelet. Since it includes a set of transforms, each with a different set of wavelet basis functions and different mother wavelets can be utilized to decompose data. The main idea of the transform is to calculate the degree of relationship between wavelets and data at different scales. Prior studies [38–40] showed that wavelet transform is suitable for analyzing non-stationary data such as Internet traffic data. The DWT produces multiple levels of frequency components by splitting the data at each level into high and low frequencies, denoting detail and approximate coefficients, respectively. The coefficients present temporal information of the data with different scales. Since the detail coefficients can detect rapid changes in the data, they are commonly used to identify discontinuity or sudden changes. Since choosing a proper wavelet function (i.e., mother wavelet) that matches well with data can maximize the correlation between the data and the mother wavelet [41,42], selecting an appropriate mother wavelet is vital to produce good performances. In network anomaly detection analysis, various DWT decomposition levels ($\gamma$) can be considered because different results can be produced depending on the applied decomposition level [43]. If a high decomposition level is used, it requires more computational costs to determine wavelet coefficients. Similarly, if a smaller decomposition level is applied, a short execution time might be needed. However, in such a case (having a smaller decomposition level), it is not easy to discover detailed internal structures. In our previous study [44], we conducted an analysis to determine an optimal decomposition level and found that $\gamma = 3$ is great for determining attack events in the NSLKDD dataset. Therefore, we utilized the same decomposition level in this study to analyze the datasets. With the decomposition level ($\gamma = 3$), various levels of detail coefficients (detail level $1 \sim \gamma$) and approximate coefficients were measured. Since statistical validation is critical to filter unnecessary coefficients, we performed a statistical validation using ANOVA to select significant features with the *p*-values ($p < 0.0001$). That is, $\omega_{i,j} = \{\omega_{1,j}, \omega_{2,j}, \cdots, \omega_{i,j}\}$ represents wavelet coefficients at the *j*th level (i.e., $j = 1, 2, \cdots, (gamma + 1)$), and *i* is the

length of the coefficients of the $j$th level. The set $\Omega = \{\omega_{k,j}\}, k = 1, 2, \cdots, m$ is the selected coefficients with the ANOVA test, and the $m$ indicates the length of the selected features.

For analyzing network traffic data, the utilization of sliding windows is often considered [45,46]. It runs with two user-defined parameters: window size ($\alpha$) and step size ($\beta$). Finding optimal window and step size is important for understanding the characteristics of the data. Different scales of feature vectors are generated depending on the window and step size. If a small window size is chosen, large feature vectors are usually generated. The step size is also important because it has a direct impact on identifying anomalous instances. A popular approach to determining the sliding window size is utilizing the time information of events or data instances. In addition, PCA is utilized to generate visualizations. It performs eigendecomposition to find the variances and coefficients of the input data by finding eigenvectors and eigenvalues. The eigenvector with the highest eigenvalue indicates the most dominant principal components representing the most vital relationship among variables. PCA is commonly used in the visualization community to represent high-dimensional data in a lower-dimensional space (either 2D or 3D space) [47,48].

## 5. Classification Analysis and Visualization

To understand the effectiveness of MRA with visualization in network traffic data analysis, classification analysis was performed. In detail, Support vector machine (SVM), Naive Bayes (NB), and k-nearest neighbors (KNN) algorithms were used for performance comparison. SVM is a classification technique that constructs a separating hyperplane. Because of its effectiveness in classifying data, SVM has been used in numerous fields such as health [49], pattern recognition [50], and network traffic data analysis [51–54]. KNN is a non-parametric supervised machine learning method that finds k nearest observations by calculating the distance between one observation and its k-nearest neighbors [55,56]. KNN generally requires less computational time to predict the output, and it has been used in the applications such as health science [57,58] and network traffic analysis [59,60]. NB uses prior and posterior probabilities to classify data. It uses the relationships among attributes to train a model. Although several assumptions need to be defined in training (such as the data in each attribute following Gaussian distribution), it has been widely used because of strong performances in analyzing data in various domains such as bioinformatics [61], health science [62], network traffic analysis [63,64].

To represent network traffic data, a coordinated multi-view (CMV) framework [65] is used to support multiple data analyses with different visualizations. The visualization includes two views, PCA Projection View and Data View. In the PCA Projection View, the first and second principal components are used to represent all network traffic data within a 2D display space (mapped to $x$- and $y$-axis, respectively). Each network event is represented as a rectangular-shaped glyph, and blue and red color attributes are used to indicate normal and attack events. It supports multiple user interaction techniques such as navigation, selection, and manipulation to help the user conduct visual analysis of the displayed network traffic data. Navigation techniques (including semantic zooming and panning) help the user navigate the PCA projection space freely. The Data View is designed to display two connected sub-views to represent normal and attack events separately by utilizing parallel coordinates. The parallel coordinates visualization [66]) shows the actual feature space of the network traffic data. In the view, each line indicates an individual network event. Therefore, a highly cluttered visual representation indicates that there are high variances in the data.

Figure 1 presents visualizations of the different intrusion detection datasets using the original raw features (Figure 1A,C) and the selected DWT features (Figure 1D,F. The visualizations with the NSLKDD dataset (Figure 1A) and the Kyoto2006 dataset (Figure 1B) show minor differences between normal and attack events. Since the overall number of attack events in the Kyoto2006 dataset is very large compared to the normal events, they show a dispersion pattern in the PCA projection space. However, the visualization with the CICIDS2017 dataset (Figure 1C) presents a significant difference since it is an imbalanced

dataset (only 12.5% indicates attack events). Because of the high similarity between normal and attack events, the attack events are not visible clearly in the PCA projection space. Figure 1D,F represent the visualizations with the DWT features. Differences between the raw and the DWT features are clear when comparing their PCA projections. The visualization with the NSLKDD dataset shows that a separated cluster representing normal events only appears in the bottom right corner (see Figure 1D). For the Kyoto2006 dataset, the difference between the two visualizations (Figure 1B,E) was not clear since the representations were highly cluttered. However, the Data View clearly shows a difference between normal and attack events. With the DWT features, the attack events become more visible in the CICIDS 2017 dataset on the right center of the PCA projection space (see red-colored attack events in Figure 1F). We provide a more detailed explanation of the generated visualizations with the DWT features in Section 6.

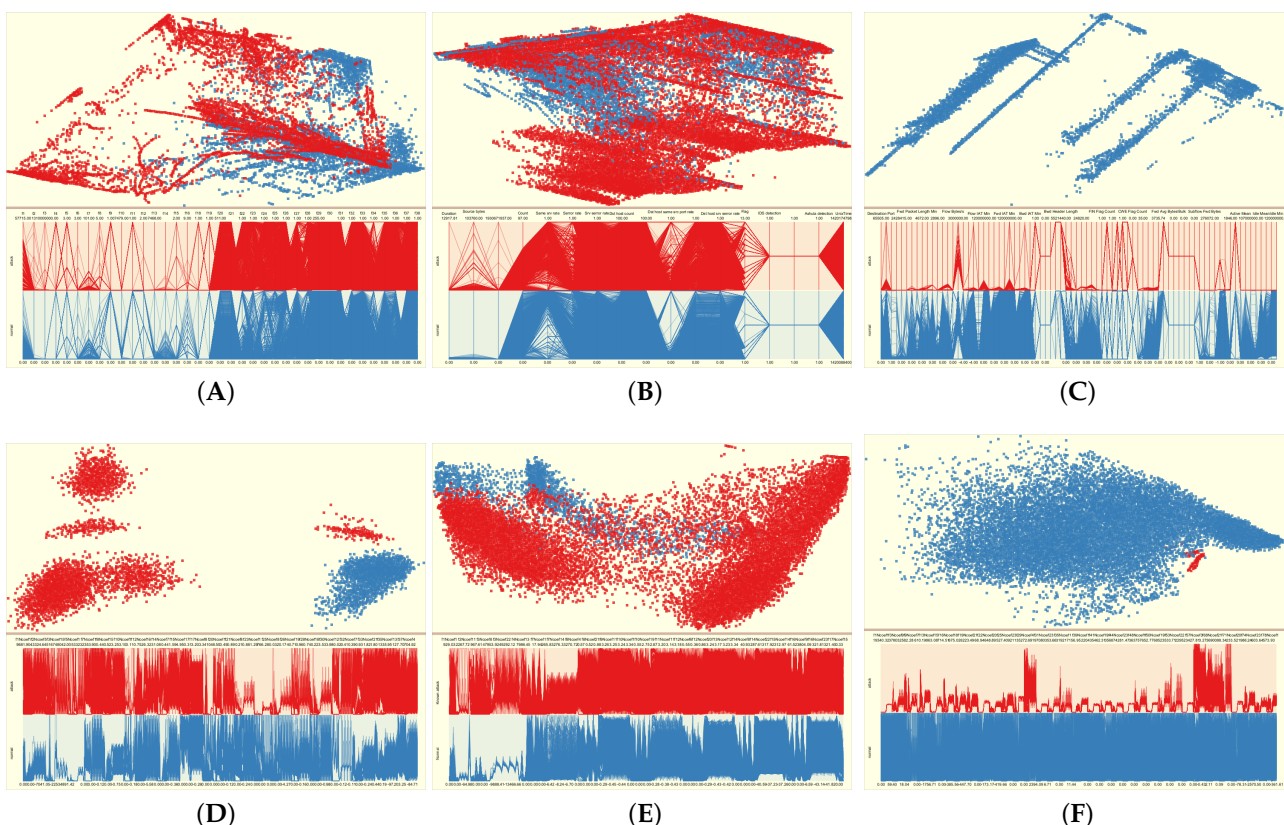

**Figure 1.** Visual representations of intrusion detection datasets. (**A–C**) represent the original raw features of the datasets. (**D–F**) show the analyzed the DWT features ($\alpha = 150$ and $\beta = 10$). All network traffic instances are mapped to blue- and red-colored glyphs to represent normal and attack events, respectively. (**A**) NSLKDD Raw Features; (**B**) Kyoto2006 Raw Features; (**C**) CICIDS2017 Raw Features; (**D**) NSLKDD DWT Features; (**E**) Kyoto2006 DWT Features; (**F**) CICIDS2017 DWT Features.

With the user interaction techniques (i.e., selection and manipulation), the user can select and manipulate network instances to understand the network traffic data within the visualization. In detail, with the selection technique, interesting network patterns can be selected or highlighted by the user. The manipulation technique assists the user in controlling the visual representation of the data. For instance, the user can select and eliminate unwanted data instances from the projection. This approach is essential in MRA with visualization because it helps the user understand the overall contribution of the removed data instances compared to the rest of the data. This feature also leads to an outlier detection analysis, as minimally contributing instances can become possible outliers.

During the visual analysis, similar network traffic instances can be identified using similarity measurements: Cosine similarity, Euclidean distance, extended Jaccard coefficient, and Pearson correlation coefficient. With the user-selected data item(s), statistically similar network traffic instances ($p < 0.05$) are determined. It helps the user initiate an analysis of the selected network traffic instances and discover possible outliers if no similar events are detected. Additionally, the user can select $k$ clusters with a hierarchical clustering method based on distance measures, such as Euclidean distance ($L^2$), Chebyshev distance ($L^\infty$), City-block distance ($L^1$), and Pearson correlation coefficient ($R^2$). This approach of identifying possible clusters may not be optimal in intrusion detection analysis due to the randomness in network traffic patterns [67]. However, understanding possible clusters is vital in MRA because it supports identifying different feature spaces through various DWT wavelet families and parameters.

## 6. Results and Discussion

When applying the DWT with various wavelet functions and levels, different numbers of instances and features are generated. Statistical validation is performed to determine significant features ($p < 0.01$) and reduce computational time while maintaining high accuracy. For analyzing the NSLKDD dataset with Daubechies 3 (db3) with $\alpha = 150$ and $\beta = 30$, a total of 4929 data instances with 572 features were determined. Under the same parameter setting with Daubechies 5 (db5), we found the same data instances (i.e., 4929) with a much larger number of features (i.e., 667). Similar results were also observed under the same parameter setting in Kyoto2006 (db3: instances = 12,694 and features = 292/db5: instances = 12,694 and features = 334) and CICIDS2017 (db3: instances = 14,850 and features = 1058/db5: instances = 14,850 and features = 1439). Figure 2 shows the average number of features and instances generated with different wavelets, window ($\alpha$), and step ($\beta$) sizes. As the window size increases, the overall number of significant DWT features increases as well (see Figure 2A–C). However, only minor changes in the number of features were observed with increased step sizes.

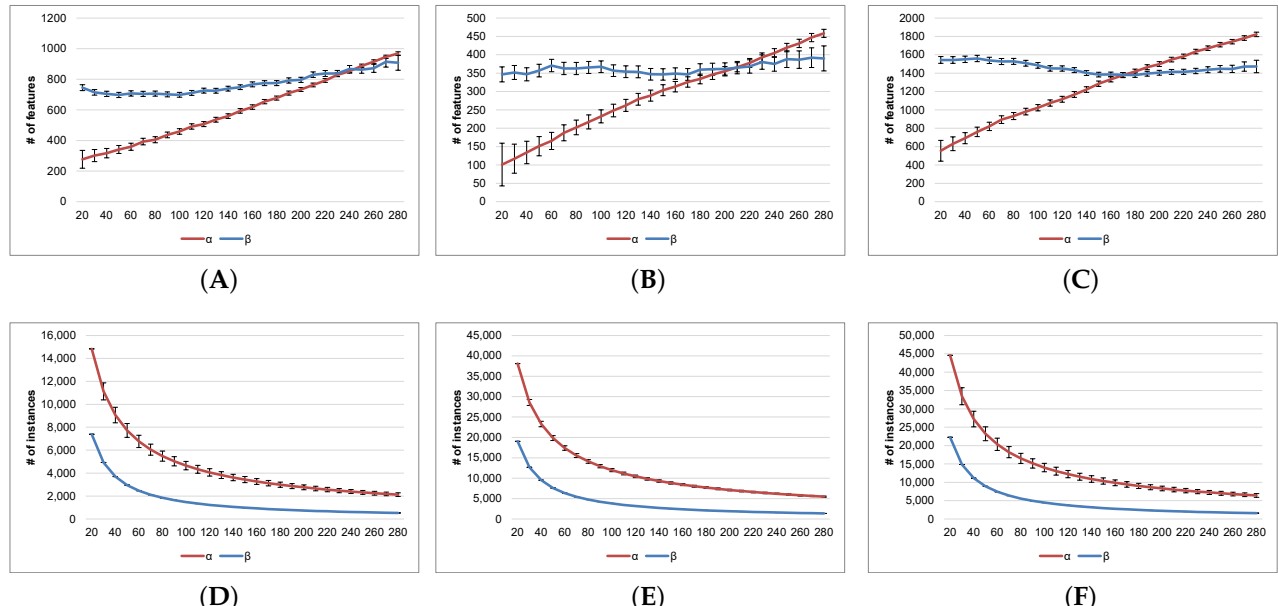

**Figure 2.** The number of determined DWT attributes and instances $\pm$ SEM depending on the window ($\alpha$) and step ($\beta$) sizes. *x*-axis indicates the size of either $\alpha$ or $\beta$. (**A**) NSLKDD; (**B**) Kyoto2006; (**C**) CICIDS2017; (**D**) NSLKDD; (**E**) Kyoto2006; (**F**) CICIDS2017.

One interesting result we found with the NSLKDD dataset was that the number of features showed a decreasing pattern when the step size increased up to 100. After the value of 100, the number of features started to increase slowly. We found a fluctuation

in the number of features with increased window sizes in the Kyoto2016 dataset. For the CICIDS2017 dataset, we identified a gradual change (slowly increased and decreased pattern) as the window size increased. On the other hand, when the step size was increased, we found a continuously increasing pattern for all datasets. We also found that there was a high variation in the number of features, especially when the step size was small (e.g., $\beta = 20$). This would be because the determined number of features might have a high variance depending on the applied wavelet function. For example, coif5 produces more features than coif1 or coif3 as the wavelet function is stretched to capture multiple frequency components at different locations of the data. By evaluating the number of instances, we determined that the size of the instances decreased if either window or step size decreased. With a small window and step size, numerous features were generated and determined as significant features. We determined that using a small step size would be beneficial for analyzing the data because it generated a large number of instances, which eventually might help train and build predictive models for detecting network attacks.

We also performed an extensive evaluation of the datasets to determine optimal parameters (i.e., sliding window and step sizes) for analyzing the data with visualization. Various window ($\alpha$) and step ($\beta$) sizes ranging from 20 to 300 and 10 to 290, respectively, were tested to examine the effect of different window and step values on the classification performances. For a fair comparison, the evaluation was managed under the condition of having a fixed value for the sliding window or the step size. Figure 3 shows visualizations with different window sizes under the same step size $\beta = 10$. We found that when a small window size was used (i.e., $\alpha <= 100$), it was difficult to separate normal and attack events clearly. For instance, most normal and attack events lay within the same local region (see Figure 3G). Similarly, we did not observe a clear separation in the CICIDS2017 dataset (see Figure 3H–L). One interesting result was that all attack events appear at the bottom of the projection space. For the NSLKDD dataset, we identified a high overlap between normal and attack events (see Figure 3A). Davies–Bouldin index (DBI) scores were also measured to determine the average similarity between clusters. The lower score closer to zero indicates the clusters are well separated. For the NSLKDD dataset, we determined that window size ($\alpha = 150$) was good for differentiating normal and attack events (see Figure 3D). However, for the Kyoto2016 and CICIDS2017 datasets, we identified different window sizes as optimal values ($\alpha = 50$ and 20), respectively. The visual representations did not show separated clusters (see Figure 3H,M). This might happen because there is a high similarity between normal and attack events in the Kyoto2016 dataset. However, for the CICIDS2017, we found that clearly separated clusters become visible when a higher window size ($\alpha > 150$) is used.

From this evaluation, we concluded that the distinction between normal and attack events became clearer when the window size increased. We also found that distinctively separated clusters could become visible because of the decreased number of instances caused by the increased window size. This might be related to the data aggregation (i.e., causing a reduced number of network instances) caused by the DWT decomposition. Understanding the phenomenon of highly decomposed data to analyze the intrusion detection datasets is essential, but it is not a primary research topic of this study. Therefore, we leave this as a possible future work. Instead, we conducted a performance evaluation study with various classification algorithms based on the window and step sizes. This helps us determine the most efficient MRA approach in analyzing intrusion detection datasets.

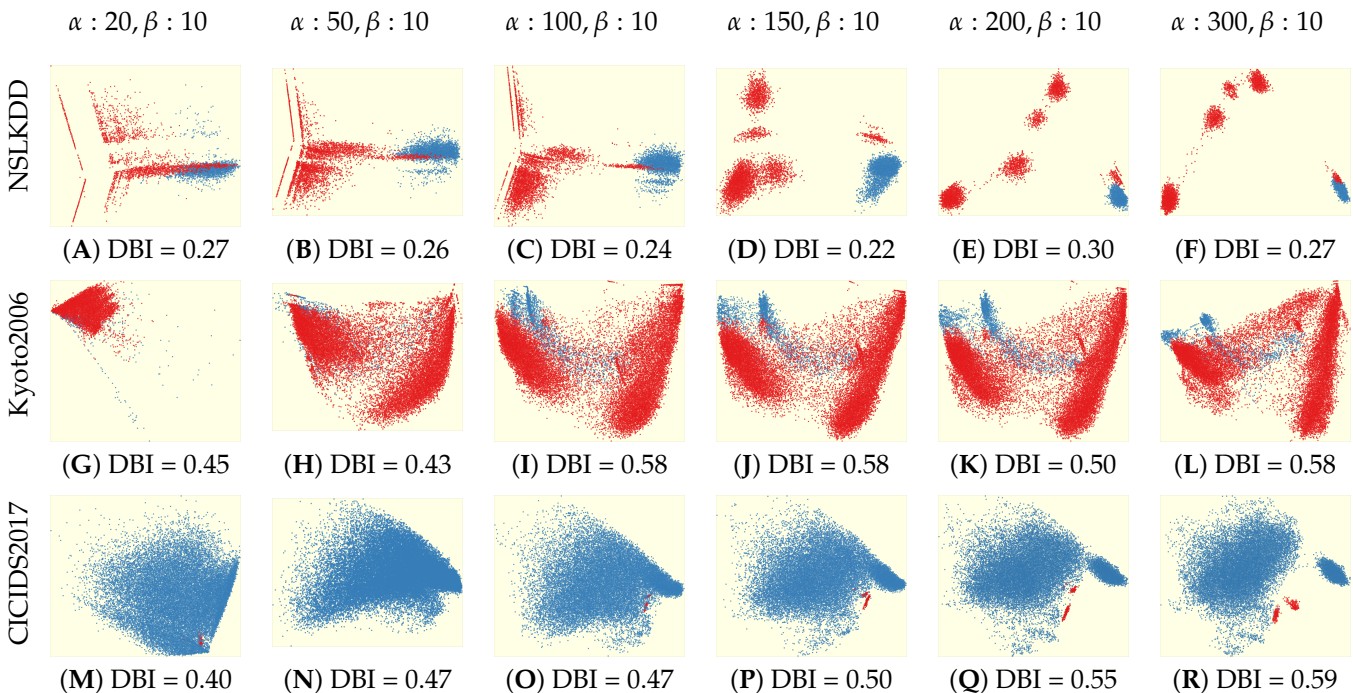

**Figure 3.** Visual representations of the conducted empirical study of identifying optimal parameters with different sliding window sizes ($\alpha$) with the wavelet function (Daubechies 3) to detect network intrusions. DBI indicates the Davies–Bouldin Index.

With the original network traffic data (using raw features), it is not easy to determine normal vs. attack activities. As shown in Figure 1, differentiating normal and attack network activities was not easy with the raw features because the activities are spread all over the places in the projection space, which make the user difficult to understand. When applying DWT wavelets, a clear separation between normal and attack activities is appeared by forming distinctive clusters. However, it is still not clear which DWT wavelets (i.e., wavelet families) produce more significant features to detect intrusions. Therefore, we tested various wavelets, such as Haar (haar), Biorthogonal (bior), Daubechies (db), Coiflets (coif), and Symlets (sym), and compared their performances. Figure 4 shows PCA projections by applying different DWT wavelets with the same window and step sizes ($\alpha = 150$ and $\beta = 10$). Depending on the characteristics of the dataset, some differences were observed. The projections with the NSLKDD dataset showed similar results when applying different wavelets. For instance, bior1.1, bior2.2, coif1, db3, and sym4 wavelets have similar visual representations. The wavelets of coif3, coif5, and sym8, as well as db5 and sym6, also resulted in similar visualizations. Interestingly, we found that Haar (haar) wavelet displayed a completely different visualization with a clearly separated cluster of normal network events. However, when analyzing the Kyoto2006 dataset, we found similar visualization results even if different wavelets were applied. We suspect it is because the Kyoto2016 dataset had different characteristics compared to the other two datasets. More specifically, the Kyoto2016 dataset was generated in a honeypot environment. A honeypot is a network-attached system used as a controlled decoy environment to examine attackers' behaviors and attack patterns. Therefore, there might be little difference between normal and attack events because all incoming network traffic to the honeypot environment is considered possible attacks. In fact, numerous network traffic events were captured as intrusive traffics. For the CICIDS2017 dataset, we also found similar visualization results. However, for evaluating the projections of attack events in different visualization results, we found minor differences. Specifically, the wavelets of coif3, coif5, sym6, and sym8 resulted in two separate clusters (see red-colored glyphs).

Figure 5 presents detailed attack information for each dataset with the db3 wavelet. DoS, Probe, and R2L attacks in the NSLKDD dataset form distinctive clusters (see Figure 5A). For the Kyoto2006 dataset, there were two visible vertically-shaped attack clusters positioned on each side in the projection space (see Figure 5B). We suspect that certain types of network attacks with two distinctive characteristics might affect forming the shape of the clusters. However, we could not further analyze the clusters due to a lack of information about the attacks. From the visualization, we also could not determine a clear distinction between normal and attack events as they appeared all over the projection space. As mentioned earlier, most network traffic events captured in the honeypot environment are considered possible attacks. Even though some network traffic events are determined as normal activities by an intrusion detection system, they could be possible unknown attack events because of the high similarity between normal and attack events. In the CICIDS2017 dataset, two different types of Brute Force attack scenarios, SSH-Patator and FTP-Patator, were tested on the Tuesday dataset. It was difficult to determine the attacks when using the raw features in the PCA projection space (see Figure 1C). However, with the DWT features, the difference between normal and attack events became visible by forming separated clusters (see Figure 5C).

Huang et al. [42] showed that Coiflet and Mexican Hat wavelets would be suitable for detecting anomalies using a five-minute, sixty-sample window. They performed the wavelet analysis on the MIT intrusion datasets and found that utilizing only the first and second coefficients might be sufficient for analyzing the network intrusions. Our study is different because we performed an in-depth analysis to identify multiple levels of wavelet coefficients for intrusion detection analysis. Classification analysis was also performed to understand the effectiveness of the various wavelets with different parameters computationally. Previously, we determined the importance of using the wavelet features compared to raw features of network traffic data [4,39]. Thus, in this paper, our classification analysis focused on understanding the effectiveness of using DWT features in visualization.

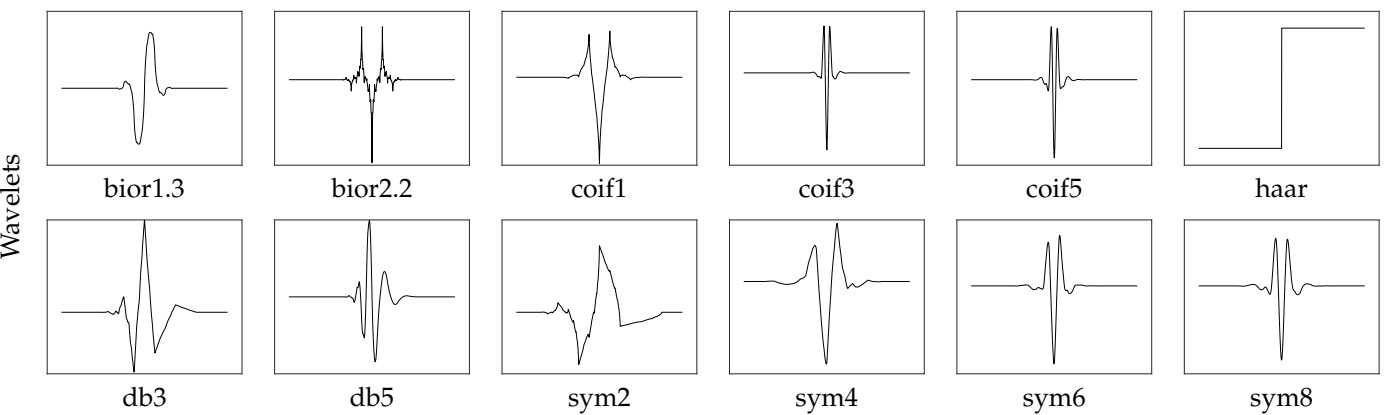

**Figure 4.** *Cont.*

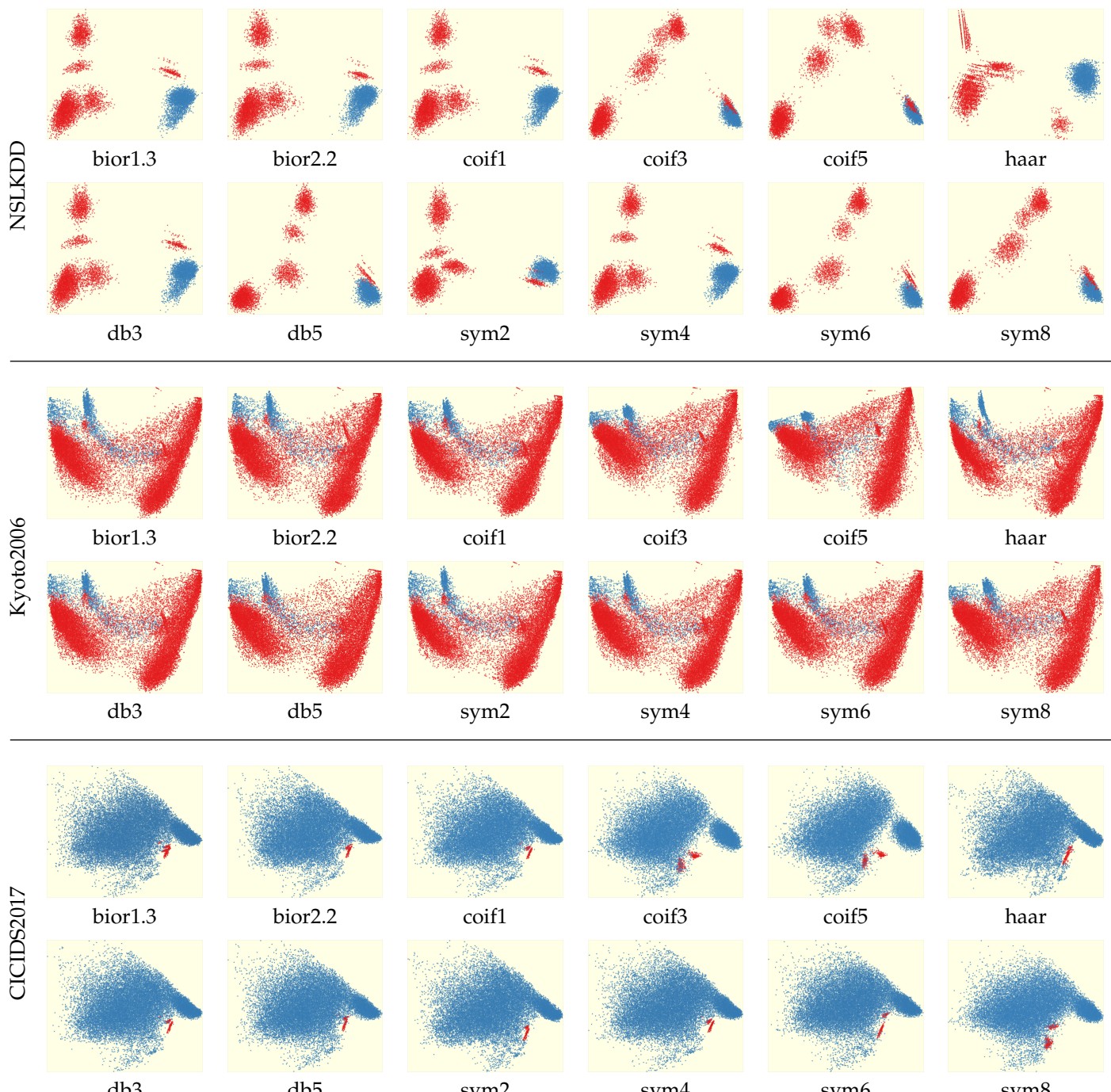

**Figure 4.** PCA projections of the DWT features ($\alpha = 150$ and $\beta = 10$) by utilizing different wavelets—Biorthogonal (bior), Coiflets (coif), Daubechies (db), and Symlets (sym).

Table 1 shows a performance comparison with different classification algorithms. It presents comparative classification results between DWT and PCA features. For the DWT features, all features are utilized. Since MRA on visualization is performed within a PCA projection space, the first two principal components are used to determine the classification performance differences. We used different metrics (i.e., accuracy, precision, recall, and F1 score) to measure performance. The DWT features showed slightly higher classification performances than the PCA features. Overall, similar results were determined except for using NB. Performance degradation was found with NB for the NSLKDD dataset, especially when using the PCA features. For the CICIDS2017 dataset, there was no

significant difference between the DWT and the PCA features, but high-performance results were observed because of data imbalance. Similarly, a high recall score was determined when analyzing the Kyoto2006 dataset with SVM because the number of attack events is relatively small.

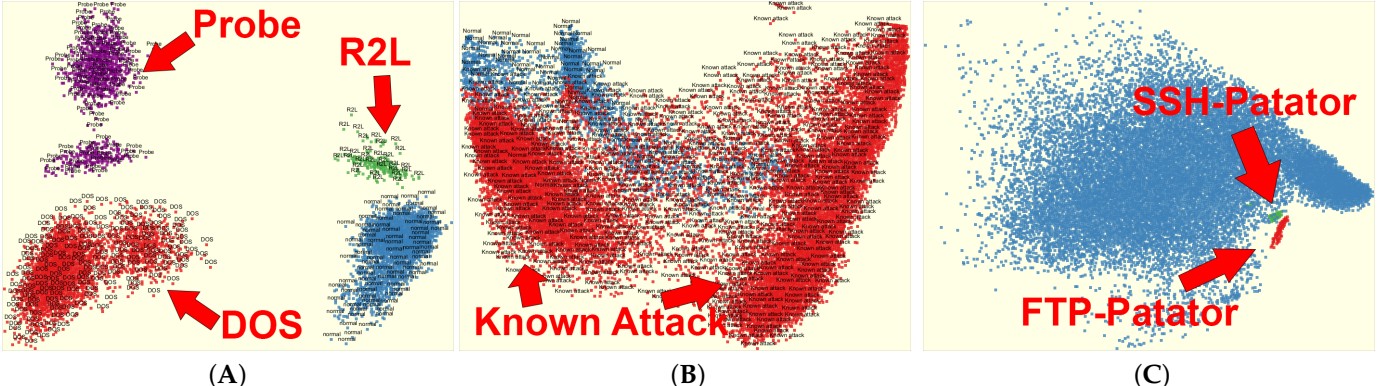

**Figure 5.** Detailed attack information with the DWT features (db3, $\alpha = 150$ and $\beta = 10$) in the PCA projections. (**A**) NSLKDD; (**B**) Kyoto2006; (**C**) CICIDS2017.

**Table 1.** Performance differences between all DWT features and PCA features with multiple classification algorithms. For the PCA features, the first two principal components are used to run the classification.

| | | DWT Features | | | PCA Features | | |
|---|---|---|---|---|---|---|---|
| Metrics | Datasets | SVM | NB | KNN | SVM | NB | KNN |
| | NSLKDD | 0.521 | 0.987 | 0.951 | 0.520 | 0.528 | 0.938 |
| Accuracy | Kyoto2006 | 0.892 | 0.632 | 0.971 | 0.893 | 0.892 | 0.951 |
| | CICIDS2017 | 0.970 | 0.998 | 0.998 | 0.970 | 0.970 | 0.994 |
| | NSLKDD | 0.521 | 0.987 | 0.951 | 0.520 | 0.528 | 0.938 |
| Precision | Kyoto2006 | 0.892 | 0.969 | 0.984 | 0.893 | 0.892 | 0.968 |
| | CICIDS2017 | 0.940 | 0.998 | 0.998 | 0.940 | 0.940 | 0.994 |
| | NSLKDD | 0.521 | 0.987 | 0.951 | 0.520 | 0.528 | 0.938 |
| Recall | Kyoto2006 | 0.998 | 0.621 | 0.984 | 0.998 | 0.999 | 0.978 |
| | CICIDS2017 | 0.970 | 0.998 | 0.998 | 0.970 | 0.970 | 0.994 |
| | NSLKDD | 0.521 | 0.987 | 0.951 | 0.520 | 0.528 | 0.938 |
| F1 score | Kyoto2006 | 0.775 | 0.916 | 0.974 | 0.943 | 0.943 | 0.973 |
| | CICIDS2017 | 0.955 | 0.998 | 0.998 | 0.955 | 0.955 | 0.994 |

From the extensive evaluation of the DWT features with various window and step sizes, we found a performance difference depending on the datasets. Figure 6 represents average F1 scores of classification algorithms based on different window and step sizes. Because of the highly imbalanced dataset, specifically the CICIDS2017 dataset, we could not find a major difference in the F1 scores with the DWT and PCA features (see orange-colored polylines). For the Kyoto2006 dataset, we found a distinguishable result. With the DWT features, we observed the average F1 scores of $0.86 \pm 0.02$. However, with the PCA features, the average F1 scores jumped to $0.95 \pm 0.002$. It is because the normal and attack events in the Kyoto2006 dataset maintain high similarities even if MRA has been applied. With PCA, similar characteristics were removed as it only determined highly significant attributes from the data. This finding indicates that PCA would be a good technique for determining

attack events from the network traffic data, especially when the attack events maintain highly similar characteristics compared to the normal events. For the NSLKDD dataset, we found lowered F1 scores with the PCA features than with the DWT features. As explained above, the DWT features are good for differentiating the normal and attack events in the NSLKDD dataset. However, this result explains that applying PCA does not benefit the performance in classifying them (normal vs. attack) if the difference between them has already been discovered through wavelet transformation.

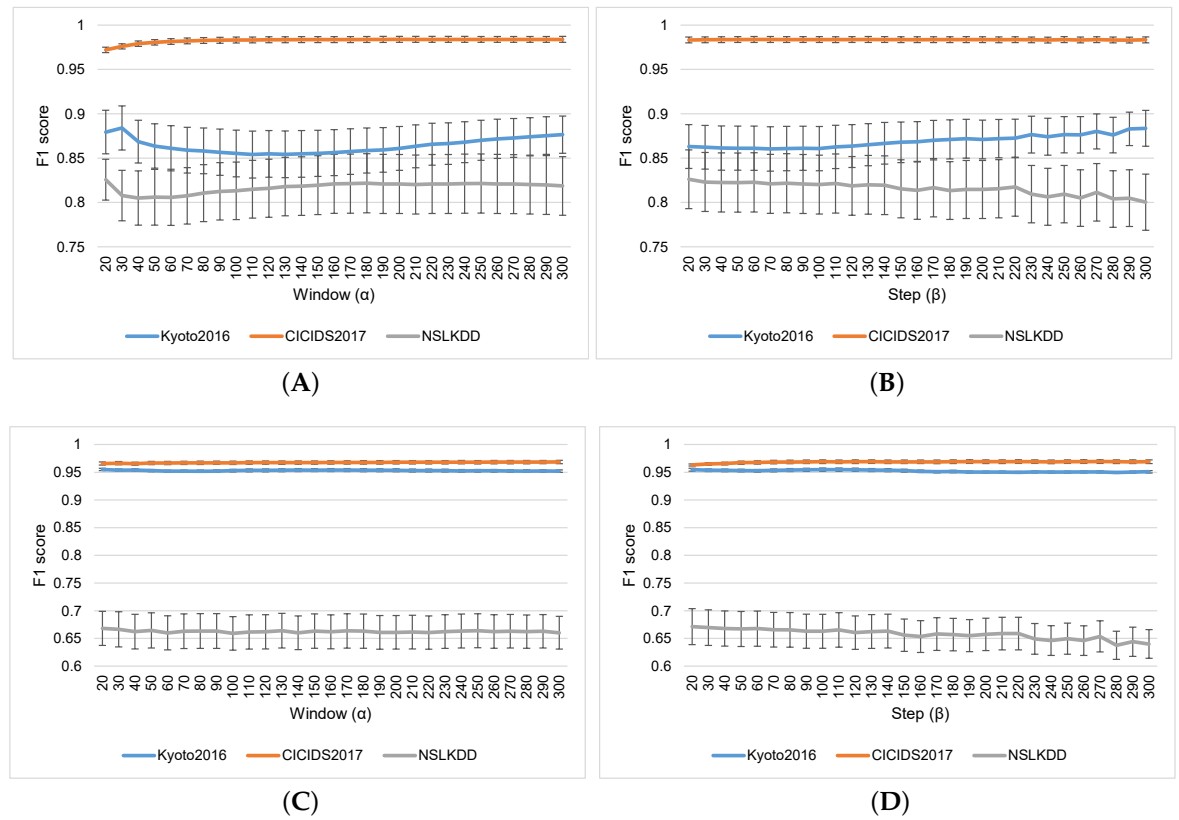

**Figure 6.** Average F1 scores $\pm$ SEM for all classification algorithms with different window and step sizes. (**A**) DWT features with $\alpha$ variation; (**B**) DWT features with $\beta$ variation; (**C**) PCA features with $\alpha$ variation; (**D**) PCA features with $\beta$ variation.

Agglomeration hierarchical clustering is applied to understand the effectiveness of performing a visual analysis within the PCA projection space as a part of MRA. It aggregates data until $k$ clusters are formed. For determining similarities, different distance measurements are utilized as Euclidean distance ($L^2$), Chebyshev distance ($L^\infty$), City-block distance ($L^1$), and Pearson correlation coefficient ($R^2$). Figure 7 shows the results when the hierarchical clustering technique is applied. Clustering results are presented with solid lines in the visualizations. Since the PCA projections were performed to the statistically validated DWT features ($\alpha = 150$ and $\beta = 10$), we found good clustering results. For the NSLKDD dataset, we could not find significant differences among different distance metrics. However, for the Kyoto2016 and CICIDS2017 datasets, different results were found. These might have happened because there was no clear visual separation between normal and attack events in the visualizations. Overall, the clustering accuracy with the Pearson correlation coefficient was slightly higher than other distance metrics.

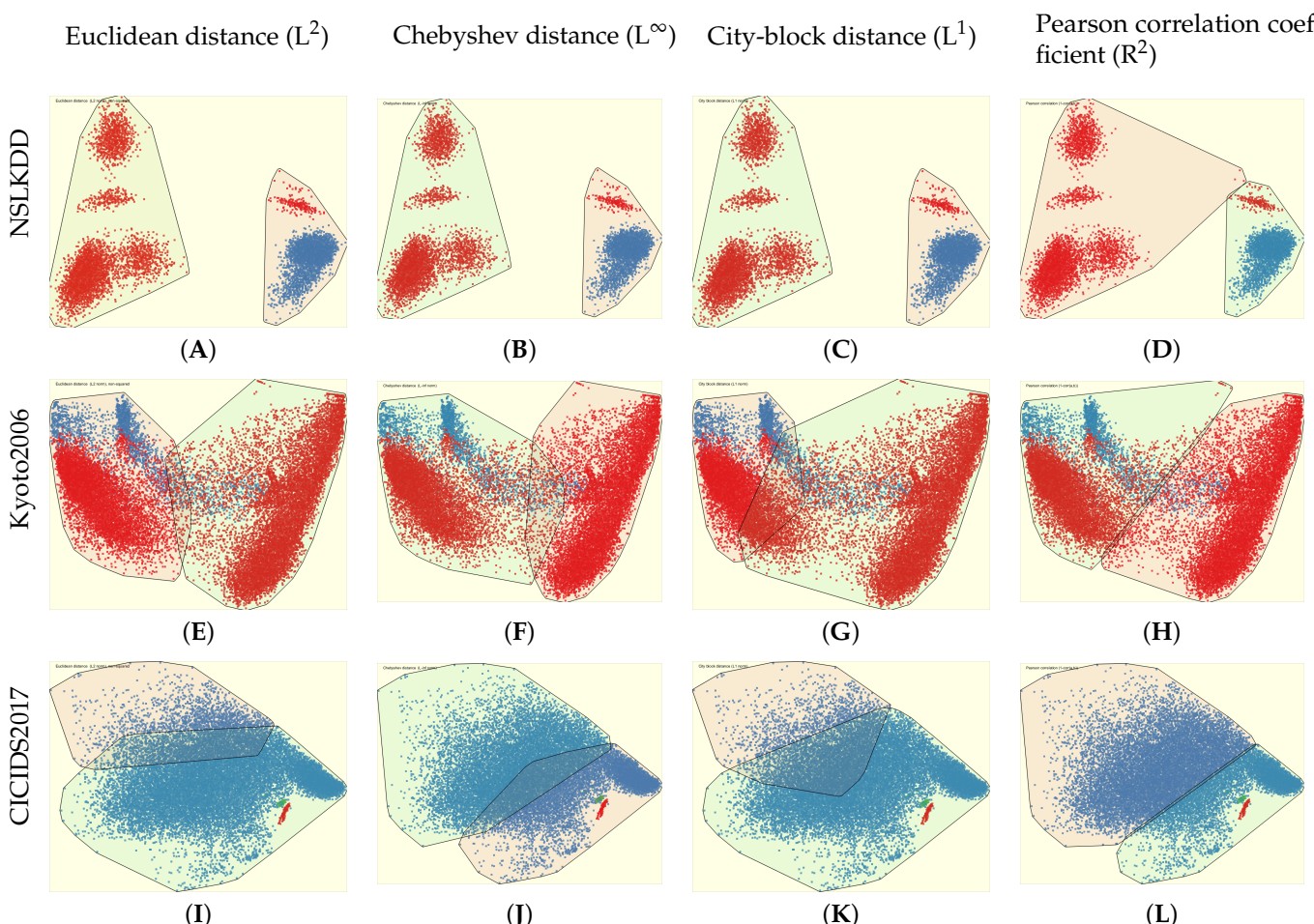

**Figure 7.** Visualizations of the analyzed $k = 2$ clusters with different distance metrics (Euclidean distance ($L^2$), Chebyshev distance ($L^\infty$), City-block distance ($L^1$), and Pearson correlation coefficient ($R^2$)) on the PCA features within visualizations using an Agglomeration hierarchical clustering technique. The determined clusters are represented as solid connected lines. (**A**) Clustering result for the NSLKDD dataset with $L^2$; (**B**) Clustering result for the NSLKDD dataset with $L^\infty$; (**C**) Clustering result for the NSLKDD dataset with $L^1$; (**D**) Clustering result for the NSLKDD dataset with $R^2$; (**E**) Clustering result for the Kyoto2006 dataset with $L^2$; (**F**) Clustering result for the Kyoto2006 dataset with $L^\infty$; (**G**) Clustering result for the Kyoto2006 dataset with $L^1$; (**H**) Clustering result for the Kyoto2006 dataset with $R^2$; (**I**) Clustering result for the CICIDS2017 dataset with $L^2$; (**J**) Clustering result for the CICIDS2017 dataset with $L^\infty$; (**K**) Clustering result for the CICIDS2017 dataset with $L^1$; (**L**) Clustering result for the CICIDS2017 dataset with $R^2$.

When applying the clustering technique, a significantly longer computational time was required to analyze the datasets (especially for the Kyoto2016 and CICIDS2017). This is because a large number of attributes are often generated depending on the sizes of wavelet window and step. As mentioned before, PCA is applied to generate visual representations by reducing the overall number of input attributes (i.e., dimensions). However, when applying PCA to determine appropriate principal components, evaluating confidence interval ($\theta$) is needed because it determines the error rate between the input and PCA projected data. For example, when mapping high-dimensional data into a low-dimensional projection space (often 2D or 3D spaces are utilized), an optimal projection space ($n$-D space) can be found by evaluating the measured eigenvectors of the covariance matrix from the input data. Of course, it is not easy to determine the optimal projection space when the number of attributes and instances is huge. In analyzing network traffic data,

minor PCA components (having smaller eigenvalues) cannot be ignored because they may include essential attributes that can be used to discover anomalies [68]. Therefore, in our visualization, the user is allowed to change the default principal components (i.e., the first two principal components) to others. This would benefit intrusion detection analysis because it helps the user analyze the data thoroughly with different PCA components.

In data classification analysis, Random Forest (RF) is often utilized. Since it builds many individual trees with bootstrapping, it has been known as a good technique for analyzing data that comes with categorical or numerical outcomes. Although it is helpful for high-dimensional applications like genomics data, especially where the number of variables exceeds the size of the observations [69], it is not suitable for analyzing highly imbalanced data. Since the intrusion detection datasets (i.e., Kyoto2006+ and CIC-IDS2017) are highly imbalanced (having a lower number of attack instances), we also found a major limitation of using RF to analyze such datasets because it produced high-performance scores (close to 0.99 or 1.0) even if different wavelet features were utilized. Thus, we excluded RF from our classification analysis. However, since it has been broadly used in network traffic analysis by researchers [64,70,71], an extensive analysis study needs to be performed to evaluate its effectiveness in analyzing highly imbalanced intrusion detection datasets.

From the evaluation studies, we determined that having a window size larger than or equal to 150 would be good for analyzing network traffic data. However, this claim needs further analysis because if the data has high similarities between normal and attack events (for example, the Kyoto2016 dataset), the distinction between them cannot easily be determined even if a higher window size is applied. As discussed above, we found that a high number of significant features were determined with increased window sizes. However, it produced fewer numbers of instances. Utilizing a relatively large window size with a small step size is highly recommended because it produces sufficient amounts of data that eventually helps analyze network traffic patterns. Additionally, an extensive evaluation must be performed to find an optimal window size. Because of the highly complex nature of network traffic data (often highly imbalanced), these findings lead us to initiate our future research in understanding the characteristics of network traffic data to determine the optimal alpha value automatically.

## 7. Conclusions and Future Works

This paper introduces a new way of analyzing intrusion detection datasets. Although various approaches have been proposed by incorporating machine learning algorithms, most techniques still suffer from detecting unknown attacks. To address the limitation, our study focuses on integrating MRA with visualization. Specifically, the DWT is used to determine significant wavelet features from the data. Then, the utilization of interactive visualization is considered to represent the network traffic features to support interactive visual analysis. Because of the complexity of network traffic data, we performed extensive evaluation studies to determine optimal parameter values for applying the DWT to different network traffic datasets. Classification analysis with SVM, NB, and KNN was performed to determine the effectiveness of using MRA with visualization. A hierarchical clustering method was also applied to identify clusters for normal and attack events by measuring the Davies–Bouldin Index scores. From the study, we found that a small step size ($\beta <= 20$) with a relatively large sliding window size ($\alpha >= 150$) formed distinctive clusters in differentiating normal and attack events. We also conclude that our approach of utilizing MRA with visualization advances network intrusion detection by generating well-separated visual clusters.

For future works, we plan to extend our study to test all possible input parameters and measure sensitivity and specificity in detecting network intrusions with various imbalanced data analysis techniques. Furthermore, a comparative study with known intrusion detection models will be conducted to determine the benefits and limitations of our approach. Since analyzing network traffic data to find optimal parameters requires tremendous com-

putational time and costs, finding a solution that automatically determines the parameters is essential. Specifically, we plan to design an approach that determines the parameters automatically by evaluating the unique characteristics of network traffic data. With the proposed approach, a real-time intrusion detection system can be designed and tested to detect intrusions in a real network environment.

**Author Contributions:** Conceptualization, D.H.J. and S.-Y.J.; methodology, D.H.J. and S.-Y.J.; software, D.H.J.; validation, D.H.J. and S.-Y.J.; formal analysis, D.H.J. and S.-Y.J.; investigation, D.H.J. and S.-Y.J.; resources, D.H.J. and S.-Y.J.; data curation, D.H.J. and S.-Y.J.; writing—original draft preparation, D.H.J., B.-K.J. and S.-Y.J.; writing—review and editing, D.H.J., B.-K.J. and S.-Y.J.; visualization, D.H.J. and S.-Y.J.; supervision, D.H.J., B.-K.J. and S.-Y.J.; project administration, D.H.J. and S.-Y.J.; funding acquisition, D.H.J. and S.-Y.J. All authors have read and agreed to the published version of the manuscript.

**Funding:** This material is based upon work supported by the National Science Foundation (Grant No. 2107451 and 2219634).

**Institutional Review Board Statement:** Not applicable.

**Informed Consent Statement:** Not applicable.

**Data Availability Statement:** The Kyoto2016+ dataset is available at the website at https://www.takakura.com/Kyoto_data/ (accessed on 10 January 2022). Both the NSLKDD dataset and the CICIDS2017 dataset were obtained at https://www.unb.ca/cic/datasets/nsl.html and https://www.unb.ca/cic/datasets/ids-2017.html, respectively (accessed on 10 January 2022). The complete analysis data and source codes will become available upon request by email.

**Conflicts of Interest:** The authors declare no conflict of interest. The funder had no role in the design of the study; in the collection, analyses, or interpretation of data; in the writing of the manuscript; or in the decision to publish the results.

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
