# Peer review of "Multi-Resolution Analysis with Visualization to Determine Network Attack Patterns"

_applsci, doi:10.3390/app13063792_

Round 1
Reviewer 1 Report
To identify the attack patterns of network traffic activities, the authors proposed a method by Multi-Resolution Analysis (MRA) and visualization, and Discrete Wavelet Transform (DWT) is utilized to extract features. The performance of Four different classification algorithms based on multiple intrusion detection datasets is presented. The topic is interesting and the paper is well organized. However, the novelty is somewhat weak and there is no solid result to demonstrate the advantages of the proposed approach. Several issues need to be addressed furthermore. Please find my detailed comments as follows:
1) The abstract needs to be improved furthermore. There is no valuable conclusion or findings in the abstract. Furthermore, please give specific experiment results to demonstrate the effectiveness or advantages of the proposed solution.
2) From the Table, it looks like DWT-based feature extraction does not show significant advantages compared with PCA based approach. Moreover, please compare the computational complexity of the two approaches. Are there advantages or disadvantages in terms of complexity?
3) There is no comparison between the proposed MRA with classical machine learning methods, such as SVM, NB, and KNN。 Readers can not see the advantage or the value of the MRA. It is better to have some comparisons.
4) It seems alpha and beta have a big impact on the performance of the approach. How to select the best alpha and beta? Is there any impact on the complexity of the proposed approach?
Author Response
Thanks for the comments. We have addressed the comments carefully. Please see the attached pdf.

Reviewer 2 Report
The title of the manuscript: to Understand network intrusion, still requires further explanation of its meaning. The data, analysis, and conclusions do not lead the content to "Understand network instruction." The author needs to reinforce the explanation in the manuscript about this meaning.
Authors must explain why MRA is used to identify unknown (or new) threats. In the previous work section, there is no introduction to the limitations of the existing solutions and their connection with the proposed use of the MRA method in this study.
In addition to the results of the research achievements that have been submitted, the author would be better if he also conveyed the limitations of the solutions offered or the results obtained in this study.
Author Response

(The authors gave the same response as above.)

Reviewer 3 Report
This manuscript introduces a way of analyzing intrusion detection datasets. To address the limitation, the authors proposed a hybrid approach that integrates computational analysis and visual analytics. The talked issue is interesting, but there are several concerns.
1) What is the main question addressed by the research?
2) What does it add to the subject area compared with other published material?
3) The conclusion should be supported by data.
Author Response

(The authors gave the same response as above.)
